# Exploration of alive-and-ventilator free days as an outcome measure for clinical trials of Resuscitative interventions

**Ari Moskowitz**[1,2]*, **Xianhong Xie**[3], **Michelle Ng Gong**[1], **Henry E. Wang**[4], **Luke Andrea**[1,2], **Yungtai Lo**[3], **Mimi Kim**[3], **for the Hospital Airway Resuscitation Trial Investigators**[¶]

**1** Division of Critical Care Medicine, Montefiore Medical Center, The Bronx, NY, United States of America, **2** Bronx Center for Critical Care Outcomes and Resuscitation Research, Montefiore Medical Center, The Bronx, NY, United States of America, **3** Department of Epidemiology and Population Health, Einstein Medical School, the Bronx, New York, United States of America, **4** Department of Emergency Medicine, The Ohio State University, Columbus, Ohio, United States of America

¶ Membership of the Termite Genome Working Group is listed in the Acknowledgments
* amoskowitz@montefiore.org

**Data Availability Statement:** There were two datasets used for this study. The first dataset is from the Pragmatic Airway Resuscitation Trial. This data is publicly available through the NHLBI

## Abstract

### Background

Outcome selection is a critically important aspect of clinical trial design. Alive-and-ventilator free days is an outcome measure commonly used in critical care clinical trials, but has not been fully explored in resuscitation science.

### Methods

A simulation study was performed to explore approaches to the definition and analysis of alive-and-ventilator free days in cardiac arrest populations. Data from an in-hospital cardiac arrest observational cohort and from the Pragmatic Airway Resuscitation Trial were used to inform and conduct the simulations and validate approaches to alive-and-ventilator free days measurement and analysis.

### Findings

Alive-and-ventilator-free days is a flexible outcome measure in cardiac arrest populations. An approach to alive-and-ventilator free days that assigns -1 days when return of spontaneous circulation is not achieved provides a wider distribution of the outcome and improves statistical power. The optimal approach to the analysis of alive-and-ventilator free days varies based on the expected impact of the intervention under study on rates of return of spontaneous circulation, survival, and ventilator-free survival.

### Conclusions

Alive-and-ventilator free days adds to the armamentarium of clinical trialists in the field of resuscitation science.

BioLINCC system (https://biolincc.nhlbi.nih.gov/home/). Data internal to our hospital system cannot be shared publicly because of health privacy protections. Data are available upon request To make inquiries contact Einstein College of Medicine Office of Human Research Affairs (irb@einsteinmed.edu).

**Funding:** Dr. Moskowitz is funded by the National Heart, Lung, and Blood Institute (R33HL162980) for his work on this study. The Pragmatic Airway Resuscitation Trial was supported by UH2/UH3-HL125163 from the National Heart, Lung, and Blood Institute. The funder had no role in study design, data collection and analysis, decision to publish, or preparation of the manuscript.

**Competing interests:** The authors have declared that no competing interests exist.

## Introduction

Outcome selection is a critically important and often complex element of clinical trial design. Selecting an outcome measure requires careful balancing of feasibility with patient-centeredness, and knowledge of the potential extent to which the outcome is modifiable by the intervention under study [1]. The International Liaison Committee on Resuscitation (ILCOR) provides a core outcome set for cardiac arrest (COSCA) [2]. This core outcome set identifies three key domains—survival, neurologic function, and health-related quality of life—and provides an outcome measure and measurement timing for each. While the COSCA provides some degree of uniformity in outcome reporting, and identifies patient-centered outcome measures, the COSCA is tailored for reporting in large, randomized clinical effectiveness trials as opposed to early phase and pilot studies.

The traditional endpoints for cardiac arrest trials are hospital survival and hospital survival with favorable neurologic outcome. Survival endpoints are clearly important but have significant drawbacks including a need for large sample sizes to achieve sufficient statistical power, an often-complex relationship to resuscitation interventions that can be muddied by care practices unrelated to study procedures, and a limited reflection of illness trajectory and recovery. Short-term outcomes, such as return of spontaneous circulation (ROSC) and survival to hospital arrival, are more proximally related to the study intervention and are often selected for clinical trials in cardiac arrest [3,4], but may not be patient-centered and do not encompass recovery. Other trials have used short-term survival to balance these potentially competing elements [5,6].

Alive-and-ventilator free days (AVFD) is an increasingly used outcome measure for clinical trials in acute respiratory failure and other critically ill populations [7–12]. AVFD is a composite outcome that includes both morbidity (days on a ventilator in those who survive) and mortality. AVFD may strike a balance between longer-term outcomes and proximal outcomes such as ROSC. AVFD carries more information than a dichotomous outcome and therefore may allow for smaller and more efficient trials [1]. Notably, the AVFD outcome not only reflects respiratory failure—as it was originally intended to do for acute respiratory distress syndrome trials—but also neurologic failure. A patient who remains comatose after cardiac arrest will likely remain mechanically ventilated until either death or some degree of neurologic recovery. While AVFD is well correlated with the modified Rankin scale [13], AVFD has not been comprehensively explored as an outcome measure for cardiac arrest clinical trials. The traditional definition of AVFD does not account for important cardiac arrest-specific outcomes such as the achievement of ROSC.

In the present study we explore AVFD as an outcome measure for cardiac arrest clinical trials. We compare the operating characteristics of different methods for defining and analyzing AVFD using simulation studies, and illustrate the approaches using data from the Pragmatic Airway Resuscitation Trial (PART) [5].

## Methods

### Study rationale and overall approach

The primary motivation for the present study was to evaluate the performance of AVFD as an outcome in a hypothetical clinical trial comparing a control and intervention strategy for advanced airway management in in-hospital cardiac arrest (IHCA). Our overall approach for assessing performance was to conduct simulation studies according to the following multi-step process: **(1)** use data from a real-world IHCA observational cohort to estimate background/control group rates of ROSC and mortality, and to characterize the distribution of

ventilator days for statistical modeling; **(2)** create five scenarios of the potential effects of an intervention on the components of AVFD that could be reasonably expected during a clinical trial; **(3)** conduct for each scenario "model-based simulations" in which data are simulated from an underlying statistical model with parameters specified to satisfy the assumptions about the control and intervention arms in steps (1) and (2); simulated data are subsequently analyzed using different statistical approaches and the operating characteristics (i.e., power and Type 1 error rate) are compared; **(4)** use the PART trial to conduct a second set of "bootstrap simulations" based on randomly drawing with replacement repeated samples of varying sizes to assess performance of the AVFD outcomes and analytic approaches under real-world settings.

## Study cohorts

An IHCA observational cohort drawn from three hospitals in an urban hospital system was used to specify the expected outcome rates in the control group for the model-based simulation studies and characterize the distribution of the AVFD data. The hospitals selected included a community hospital, a tertiary care hospital, and a quaternary care hospital. Adult patients (age ≥18 years) who suffered an in-hospital cardiac arrest without an advanced airway already in place were included. Patients with a diagnosis of COVID-19 were excluded. Patients were enrolled into the cohort between the years 2018 and 2021. A waiver of informed consent was obtained from the Einstein Institutional Review Board (# 2021–13077) given the retrospective nature of the study. All data was anonymized prior to analysis although authors maintained a separate cross-linked codesheet to allow re-identification of patients if necessary. Data were accessed on February 4th, 2022.

The PART trial, a cluster-randomized with crossover clinical trial of two different approaches to advanced airway management (laryngeal tube [LT] supraglottic airway vs. endotracheal intubation) for out-of-hospital cardiac arrest, was used to generate the data for the "real-world" bootstrap simulations and to illustrate the methods in the example. The inclusion and exclusion criteria for PART can be found in the original trial publication [5]. PART was conducted under United States federal rules for conduct of emergency research under Exception From Informed Consent (21 CFR 50.24). A fully anonymized dataset that did not allow for re-identification patients for the present analysis was received on April 5th, 2023. Data was collected using Utstein-style definitions [14]. Informational regarding the IHCA observational cohort can be found in Supplemental Table 1 and information regarding the PART cohort in the original trial publication [5].

## Outcome definitions

In the traditional measurement approach of AVFD, a patient who dies at any time between trial entry and the end of the study period (commonly 28 days) is assigned an AVFD of 0.

**Table 1. Hypothetical scenarios evaluated.**

| Scenario Number | ROSC Rate Control | ROSC Rate Intervention | 28-Day Survival Control | 28-Day Survival Intervention | AVFD in 28-day SURVIVORS Control (mean, SD) | AVFD in 28-day SURVIVORS Intervention (mean, SD) |
|---|---|---|---|---|---|---|
| 1 | 38% | 48% | 8% | 13% | 19 days (±10 days) | 21 days (±10 days) |
| 2 | 38% | 48% | 8% | 8% | 19 days (±10 days) | 19 days (±10 days) |
| 3 | 38% | 48% | 8% | 8% | 19 days (±10 days) | 17 days (±10 days) |
| 4 | 38% | 42% | 8% | 10% | 19 days (±10 days) | 20 days (±10 days) |
| 5 | 38% | 48% | 8% | 10% | 19 days (±10 days) | 20 days (±10 days) |

AVFD is generally measured using a first on/last off approach, such that intervening days where a patient may be liberated from mechanical ventilation but are subsequently placed back on mechanical ventilation are not counted as ventilator-free [7,8]. Mechanical ventilation for calculation of AVFD is defined as the administration of positive pressure through an invasive endotracheal or tracheostomy tube. Investigators have identified two potential pitfalls of this approach which have led to adjustments of the AVFD definition in some trials. First, there is no differentiation between death and ongoing mechanical ventilation at the end of the study period. This may not be patient-centered as patients may find ongoing mechanical ventilation preferable to death. Second, there may be skewing of the data towards the tails of the distribution (day 0 and day 28). One approach used in recent high-impact clinical trials is to assign death a score of -1 and to consider the outcome on an ordinal scale from -1 to 28 [15,16]. However, this does not differentiate death without achieving ROSC from death after ROSC.

In this study we explored the traditional approach of defining AVFD using a scale from 0 to 28 with all deaths = 0 (referred to as AVFD1). We additionally evaluated an approach assigning failure to achieve ROSC as -1 and death after ROSC but before 28-days as 0 (referred to as AVFD2, on a scale -1 to 28). AVFD2 considers the achievement of ROSC to be an important, cardiac-arrest specific outcome that is desirable and should be considered differently than failure to achieve ROSC. It is also important to note that when AVFD is analyzed as an ordinal or ranked outcome using a Wilcoxon rank sum test or proportional odds model (see below), the actual value assigned to failure to achieve ROSC is irrelevant as long as this outcome is given the lowest possible AVDF value. That is, assigning -1, -50, or -1000 to failure to achieve ROSC will yield identical results.

## Scenarios considered for the in-hospital cardiac arrest hypothetical clinical trial

For the model-based simulations in the setting of a hypothetical clinical trial comparing two strategies for airway management during IHCA, a number of scenarios were considered (Table 1). These scenarios were deemed to be the most plausible based on data from the IHCA observational cohort and prior experience. They also reflect a range of possible effects of a hypothetical intervention on the individual components that comprise the two AVFD definitions. These include scenarios where the intervention: 1) improves ROSC rates, 28-day survival, and ventilator-free days in survivors; 2) improves ROSC rates but has no impact on 28-day survival or ventilator-free days in survivors; 3) improves ROSC rates, has no impact on 28-day survival, and results in fewer ventilator free days in survivors (e.g. more ROSC but higher rates of aspiration and resultant pneumonia); 4) improves ROSC, 28-day survival, and ventilator free days but the effect size is smaller than in the first scenario, and 5) improves ROSC rates as in the first scenario but has a smaller impact on 28-day survival and ventilator-free days in survivors.

## Simulation studies

**Model based simulations.** For each scenario in Table 1 and sample sizes ranging from 400 to 1000, we simulated 1000 data sets. A multinomial logistic regression model was first used to randomly generate for each subject their status with respect to failure to achieve ROSC/death after ROSC/survival. For the survivors, a beta-binomial regression model was then used to randomly generate the number of alive and vent free days. This process was repeated until the desired sample size was achieved for each simulated data set. The beta-binomial model was selected over others (e.g., negative binomial) to model AVFD among survivors since it yielded the best fit to the IHCA observational data that showed an excess number of

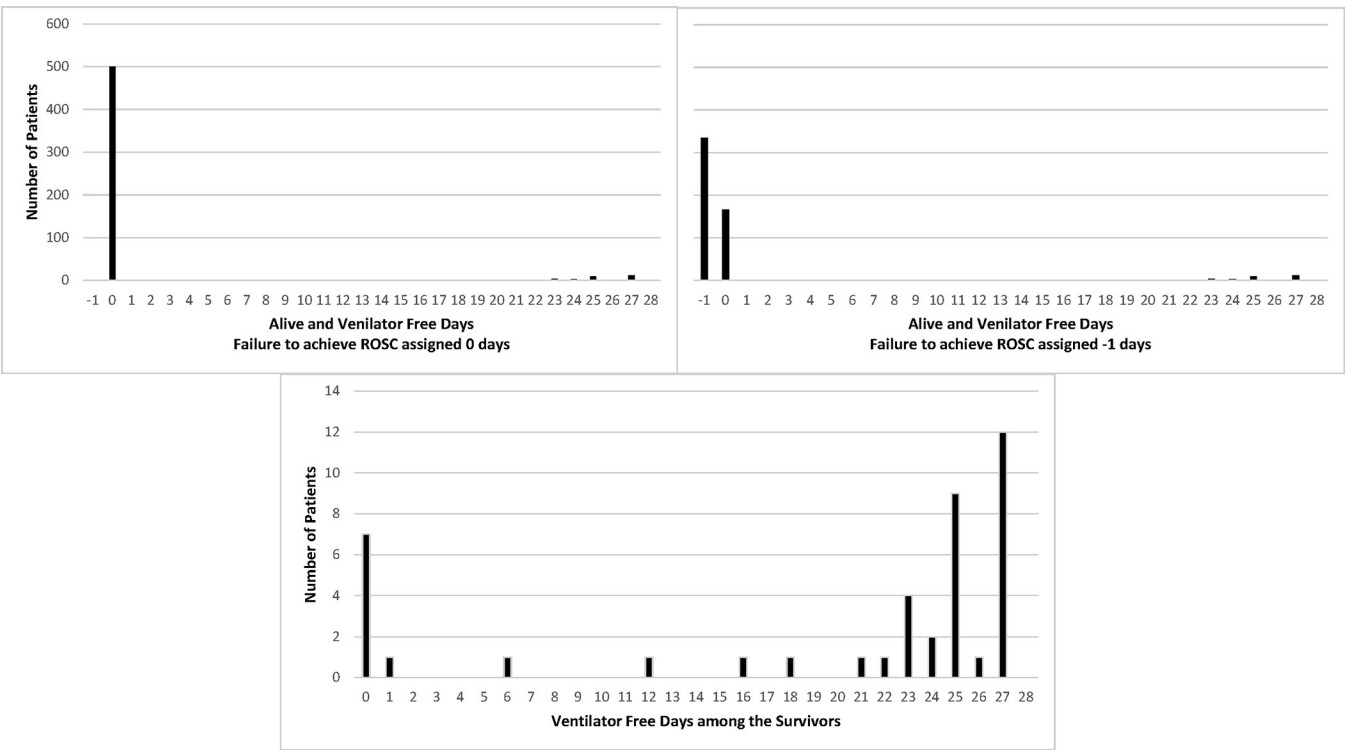

**Fig 1. Distribution of AVFD in the IHCA observational cohort.** Top left diagram with failure to achieve ROSC, death before 28-days, and persistent mechanical ventilation at 28-days all given a value of 0 AVFD. Top right diagram assigning -1 AVFD to patients who fail to achieve ROSC. Bottom diagram: ventilation free days among the survivors.

values at both ends of the distribution (Fig 1). The parameters in all regression models were specified to yield the assumed effect sizes for each scenario in Table 1. Further details about the model-based simulations are provided in the Data Supplement.

**Bootstrap data simulations.** A second set of simulations was also conducted using boot-strapped samples from the PART trial data to assess performance of the different statistical methods using data from an out-of-hospital real-world trial cohort. Data were repeatedly sampled from the PART data with replacement to generate 1000 bootstrap samples with the desired sample size.

**Statistical methods.** For each of the 1000 simulated data sets created by the model-based and bootstrap approaches, the following statistical methods were applied to analyze the intervention effect using AVFD1 and AVFD2 as the outcomes: (1) Two-sample T-test; (2) Wilcoxon rank-sum test; (3) Proportional Odds Logistic Regression Model; (4) Two-Part modeling framework. Various types of two-part models exist and have been applied in a number of different fields [17]. In our implementation of the two-part approach, the first part involves fitting a binary or multinomial logistic regression model with the categorical outcome specified either with two-levels: death/survival for AFVD1, or three-levels: failure to achieve ROSC/death after ROSC/survival for AVFD2. In the second part of the two-part method, a beta-binomial regression model is fit to the number of vent-free days among survivors. This two-part approach was considered because it offers greater flexibility than standard methods in modeling the intervention effects and hence yields more insights about the nature of the effects; this is especially important when the magnitude and direction of the intervention effects vary across the components of the composite outcome. Separate p-values and treatment

effect estimates are generated from each of the two parts; a p-value for the overall treatment effect is also obtained by performing a likelihood ratio chi-square test.

Each simulated data set was analyzed using the four statistical methods described above and power/type I error rate for each approach was computed as the proportion of the relevant simulated data sets in which the two-sided p-value for the intervention effect was less than 0.05. We additionally compared statistical power using AVFD to a binary survival outcome measure.

Finally, we provide an example using the original data from the PART trial to illustrate the analysis of AVFD1 and AVFD2 using the different statistical methods. For simplicity, it was assumed in all analyses that data from different subjects were uncorrelated.

## Results

### In-hospital cardiac arrest observational cohort

A total of 535 patients were included. The distribution of AVFDs under the traditional measurement approach, AVFD1 (0 to 28 days), and the proposed AVFD2 approach (-1 to 28 days, failure to achieve ROSC assigned -1) can be found in Fig 1. The distribution overall is not Gaussian and instead is mirrored J-shape, with a right-skew. If restricted to the survivors, it has a J-shape, with left-skew. The IHCA observation data was used to determine the expected outcomes in the control group and the statistical methods for analyzing AVFD in the simulation studies.

### Model-based simulation approach

In general, AVFD2, which differentiates death with and without ROSC, yields higher power than AVFD1 across all methods and all scenarios. Under scenario 1, power of the different methods and outcomes is shown in Fig 2. With AVFD1 (all deaths = 0), the T-test had slightly higher power than the other methods; power ranged from 72.3% with the proportional odds model to 80.1% for the T-test with N = 1000 subjects. With AVFD2 (failure to achieve ROSC = -1), the proportional odds model showed the highest power: 93.3% with n = 1000, compared to 84.7% with the T-test. The two-part approach yielded power that was in between the power of the other methods for both AVFD1 and AVFD2 (Figs 2 and 3).

Results for the remaining scenarios can be found in (S1-S8 Figs in S1 File). We note that under scenario 2, which assumes the only difference between the treatment arms is in the ROSC rate, the two treatment groups are completely equivalent with respect to AVFD1. Therefore, the simulation results in this case correspond to the Type I error rates, which are shown to be close to the target 5% with all methods (S1 Fig in S1 File). For Scenario 3, in which intervention improves ROSC but decreases AVFD in survivors, all methods perform poorly with AVFD1 (S3 Fig in S1 File). The T-test continues to perform very poorly with AVFD2 at all sample sizes; whereas the other methods attain more than 80% power with N = 1000 (S4 Fig in S1 File). The two-part model shows slightly higher power consistently across sample sizes than the proportional odds model for this scenario where the treatment effect is inconsistent across outcomes. Effect sizes are smaller in scenarios 4 and 5 so power is uniformly lower for all methods (S5-S8 Figs in S1 File). However, the proportional odds model tends to outperform the others with AVFD2. Results for the Wilcoxon and proportional odds methods were nearly identical across all scenarios.

For comparison, power using 28-day mortality as the outcome was 73.3% in scenario 1, 0.0% in scenarios 2 & 3, and 19.7% in scenarios 4 & 5 with N = 1000 subjects, which is lower than the corresponding power with AVFD2 analyzed using any of the statistical methods.

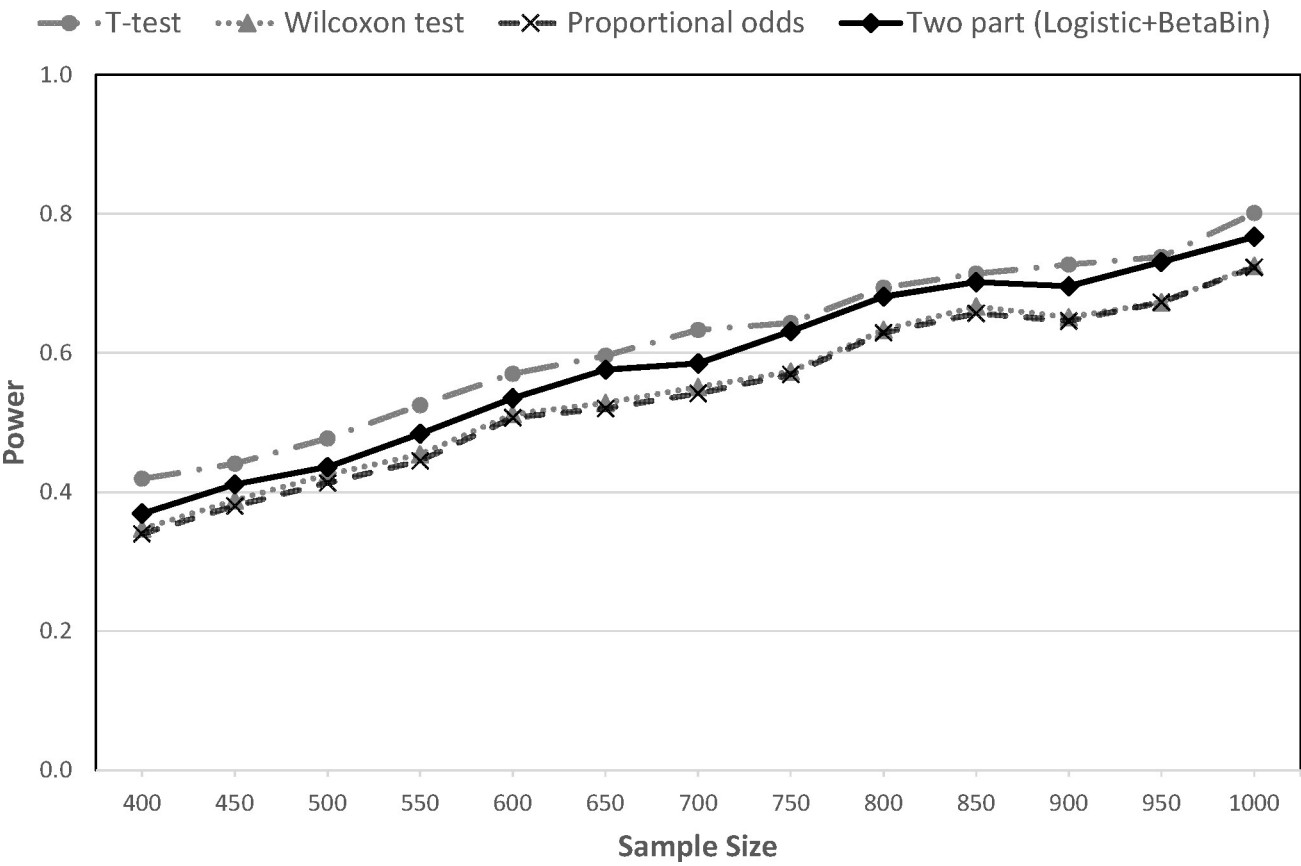

**Fig 2. Statistical power for the traditional AVFD approach in scenario 1 (AVFD1).**

### Bootstrapped simulations using pragmatic airway resuscitation trial data

With AVFD1 and simulation distributions based on bootstrapped samples from PART, the proportional odds model and T-test showed consistently higher power than the two-part method across different sample sizes; with N = 2000, power was 81%, 78%, and 66%, respectively, for the three approaches (S9 Fig in S1 File). Power for all methods exceeded 80% when N = 2800 (PART trial sample size was 3004 participants). Similar trends were observed with AVFD2, although power tended to be greater than with AVDF1, as in the model-based simulations, and differences between methods were smaller. All methods showed greater than 80% power with AVDF2 with N = 2300 (S10 Fig in S1 File).

### Example: Pragmatic airway resuscitation trial (Out-of-hospital arrest) cohort

In PART 3,004 patients were randomized to LT or endotracheal intubation. Overall, 1,173 (39.1%) patients achieved ROSC, with 547 (36.6%) assigned to the intubation group and 626 (41.5%) assigned to the LT group achieving ROSC. A total of 289 (9.6%) patients survived to hospital discharge—124 (8.3%) and 165 (10.9%) in the intubation and LT groups respectively. Mean AVFD1 in survivors was 25.2 (SD 5.5) days overall, 24.8 (SD 6.7) days in the intubation group, and 25.6 (SD 4.6) days in the LT group.

Table 2 shows the results from analysis of AVFD1 and AVFD2 using different statistical methods. In all analyses, LT was shown to be significantly better than ETI. The two-part model

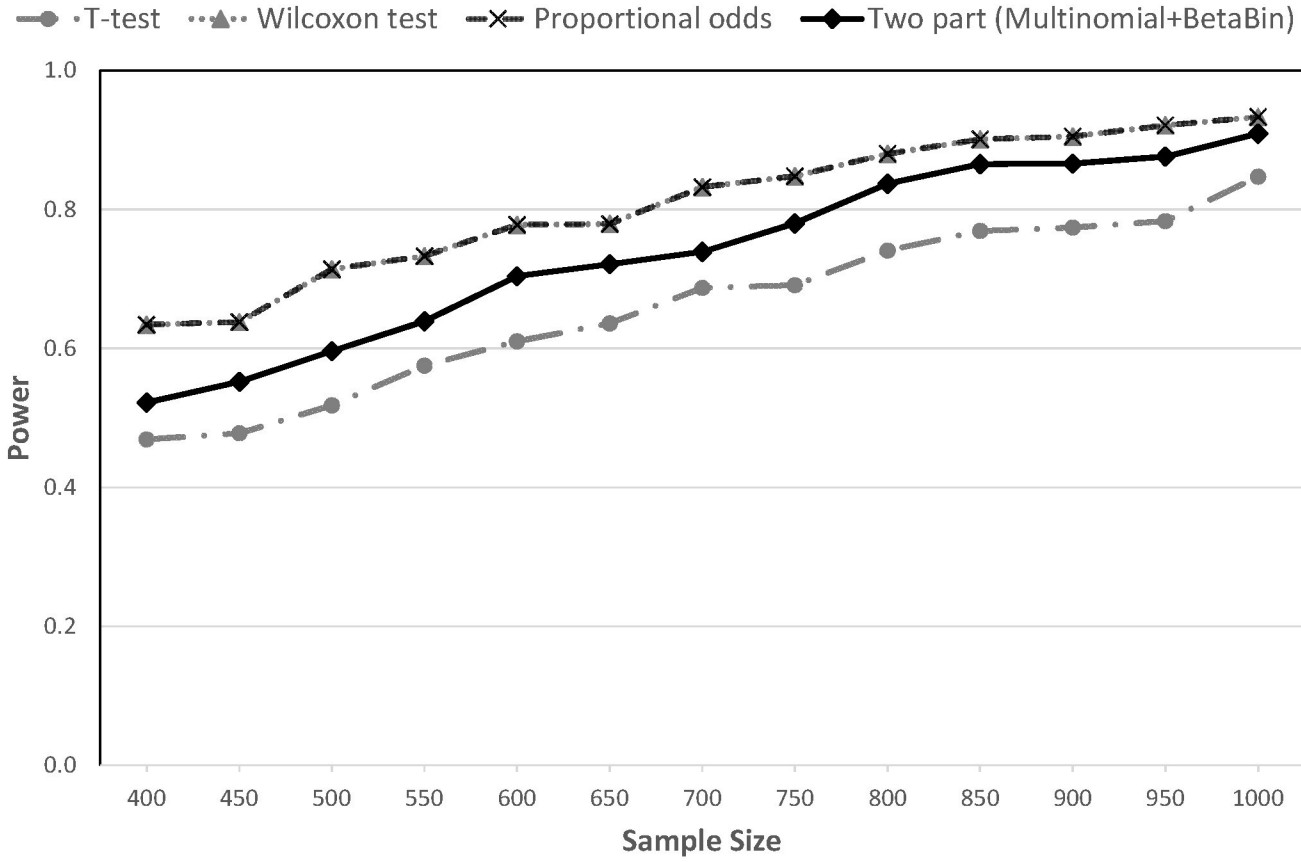

**Fig 3. Statistical power for the AVFD approach assigning -1 to patients who do not achieve ROSC in scenario 1 (AVFD2).**

provided additional insights on the intervention effects, with AVFD2 results showing that the beneficial effect of LT compared to ETI is primarily due to the reduced risk of failure to achieve ROSC (OR = 0.80; 95% CI: 0.69, 0.93; p = 0.003) and all deaths (OR = 0.65; 95% CI: 0.49,0.85; p = 0.002). LT did not appear to appreciably increase likelihood of being ventilator-free in survivors (OR = 1.10; 95% CI: 0.83, 1.46; p = 0.52); this result is the same with either AVDF1 and AVFD2 since data from survivors is not affected by how failure to achieve ROSC is coded.

## Discussion

In the present study we tested two different definitions of AVFD and found that a definition incorporating a penalty for failure to achieve ROSC had greater statistical power in most scenarios. Further, analysis of AVFD using a proportional odds model generally yields the highest power, although in circumstances where the effects of the intervention on components of the composite outcome are in different directions, a two-part model may be preferable. This study builds on recent work demonstrating correlation between AVFD and more traditional cardiac arrest outcomes [13]. This study is important as it lays the groundwork for the use of AVFD with accounting for ROSC as an outcome measure for cardiac arrest clinical trials and other studies.

Free-days outcomes (i.e. the total number of days alive and free of a supportive therapy) are increasingly used for critical care clinical trials [1,12] and for trials in other medical conditions [18–21]. While standard definitions for calculating free-days outcomes exist, there is some variation from clinical trial to clinical trial [22]. One key area of variation in recent clinical trials is

**Table 2. Pragmatic airway resuscitation trial cohort analysis.**

**AVFD1 (All deaths = 0)**

| Statistical Method | LT | ETI | Treatment Effect Estimate | P-value |
|---|---|---|---|---|
| T-test | Death within 28-days (n, %) = (1345, 90.6%)<br>AVFD in survivors to 28 days (mean, SD) = (25.57 ± 4.57) | Death within 28-days (n, %) = (1370, 93.7%)<br>AVFD in survivors to 28 days (mean, SD) = (24.75 ± 6.74) | Difference in mean AVFD1: 0.85 days (95% CI: 0.35, 1.36) | 0.001 |
| Proportional odds model | | | OR (of better outcome) = 1.62 (95% CI: 1.23, 2.14) | 0.001 |
| Two-part model | | | $OR_{death}$ = 0.65 (95% CI: 0.49, 0.85)<br>$OR_{AVFD\ survivors}$ = 1.10 (95% CI: 0.83, 1.46) | P = 0.006 (overall) P = 0.002 (death)<br>P = 0.52 (AVFD in survivors) |

**AVFD2 (Failure to achieve ROSC = -1; death after ROSC = 0)**

| Statistical Method | LT | ETI | Treatment Effect Estimate | P-value |
|---|---|---|---|---|
| T-test | ROSC (n, %) = (601, 40.5%)<br>Death within 28-days (n, %) = (1345, 90.6%)<br>AVFD in survivors to 28 days (mean, SD) = (25.57 ± 4.57) | ROSC (n, %) = (515, 35.2%)<br>Death within 28-days (n, %) = (1370, 93.7%)<br>AVFD in survivors to 28 days (mean, SD) = (24.75 ± 6.74) | Absolute difference 0.91 days (95% CI: 0.39, 1.42) | 0.001 |
| Proportional odds model | | | OR (of better outcome) = 1.29 (95% CI: 1.11, 1.49) | 0.001 |
| Two-part model | | | $OR_{death\ no\ ROSC}$ = 0.80 (95% CI: 0.69, 0.93)<br>$OR_{death\ any}$ = 0.65 (95% CI: 0.49, 0.85)<br>$OR_{AVFD\ survivors}$ = 1.10 (95% CI: 0.83, 1.46) | P = 0.003 (overall) P = 0.003 (death no ROSC)<br>P = 0.002 (any death)<br>P = 0.52 (AVFD in survivors) |

the assignment of death before the end of the study period (often 28-days) as -1 instead of 0 free-days [1,16,23]. The rationale for this is that persistent organ support at the end of the study period may be preferable to the patient as compared to death. The inclusion of -1 for death additionally widens the distribution of events and mitigates issues with clustering of outcome events at 0 days.

In the present study, we extended the above concept and explored assigning -1 to patients who did not achieve ROSC. ROSC is an outcome specific to cardiac arrest resuscitation that is commonly used as an outcome measure for cardiac arrest clinical trials. We demonstrate that assigning failure to achieve ROSC -1 generally improves the statistical power of a hypothetical clinical trial under a number of different potential scenarios. We assume that not achieving ROSC is a less desired result to patients and families than achieving ROSC followed subsequently by death. Achieving ROSC, even for patients who are unlikely to survive long post-ROSC, can allow family members a final opportunity to visit or carry out religious services. Further, after ROSC there is the possibility of organ donation for those patients who are resuscitated but then undergo donation after cardiac or brain death. Organ donation is an important consideration when examining the cost-effectiveness of cardiac arrest interventions, a key element of post-ROSC IHCA care [24,25]. The achievement of ROSC also allows for further study of post-ROSC critical care—an area in acute need of further study and outcome improvement [26]. Another option not considered in this study, but that could be analyzed in future efforts would be the assignment of additional levels for other unfavorable outcomes such as -2 AVFD for failure to achieve ROSC and -1 for achievement of ROSC followed by death before the end of the follow-up period.

We additionally explored a number of different approaches to analysis of AVFD. The proportional odds model was generally the most powerful when treatment effects on the different AVDF components (ROSC, survival, vent-free days) were all in the same direction. Notably, however, a two-part model was more powerful when treatment effects were in different directions (e.g. higher rates of ROSC but fewer ventilator-free days in initial survivors). The two-part model additionally yields greater insights about the relative effect of the intervention on

different aspects of the composite outcome. The proportional odds and two-part models that we considered yield odds ratios to quantify the magnitude of the treatment effect, which can be less interpretable than relative risks and also have the statistical limitation of non-collapsability [27]. Alternative analytic approaches that circumvent use of the odds ratio could be considered, but these also have limitations. A modified Poisson regression, for instance, may falter due to misspecification of the outcome distribution [28].

In this study, we assumed all subjects were independent (i.e., no between-subject correlation in the data) for both the IHCA and PART cohorts. The methods can be easily extended to account for the correlation in cluster randomized and cluster-randomized cross-over trials by fitting mixed effects models. We additionally note that, as with any composite outcome, it would be important to interpret the AVFD outcome within the context of individual analyses of its component parts—ROSC, mortality, and ventilator-free days. In the present study we highlight the possibility that one outcome component might improve in response to an intervention and another may worsen—individual investigators should consider this possibility *a priori* as part of analysis planning when using AVFD.

The findings and conclusions of this work have several limitations. First, while there are theoretical patient-centered advantages to the outcome of ROSC, additional efforts in patient communities are urgently needed to explore the value of ROSC to cardiac arrest victims and their families. This work is important but complex given that those who fail to achieve ROSC and those who achieve ROSC but then expire without regaining consciousness cannot be surveyed. Second, the assignment of -1 AVFD to those who fail to achieve ROSC is somewhat arbitrary. While analysis methods that treat AVFD as an ordinal outcome, such as the proportional odds model, may not be impacted by the actual values assigned to AVFD, as long as the ordering or ranks of the values are preserved—other analysis methods treating AVFD as a continuous variable may be impacted. This is another key area for patient and caregiver consultation. Third, although more severe brain injury is associated with increased need for mechanical ventilation [29], it is not clear the extent to which brain injury severity impacts AVFD. Fourth, the scenarios described are based on a local cohort and rates of ROSC, VFD, and survival may be different in other systems. Finally, the distribution of AVFD in patients who survive cardiac arrest excluding those who do not achieve ROSC is quite different [30] and the findings of this study may not directly apply.

## Conclusions

In the present study we extend the understanding of AVFD as an outcome measure for cardiac arrest clinical trials. We demonstrate that the use of an AVFD definition that separates death without ROSC from death with ROSC, increases statistical power under a number of potential scenarios. We further highlight the potential role of a two-part analysis model, which has not previously been well described in the field of critical care and resuscitation, and might have an important place in the analysis of AVFD.

## Supporting information

**S1 File. Supporting information can be found in the file "Simulation Manuscript Data Supplement_R1.docx".**
(DOCX)

## Acknowledgments

Additional Contributing Authors part of the Hospital Airway Resuscitation Trial (HART)

**Authors:** Ariel Shiloh MD[1]; John Cardasis MD[2]; Colleen Carty MS[2]; Susan McAllen RN MS[3]; David Esses MD[4]; Carlo Lutz MD[4]; Mai Takematsu MD[4]; Jose Romero MD[4]; Kristen Schimmrich MD[4]; Daniel G. Fein MD[5]; Amos Dodi MD[1]; Samuel Rednor MD[1]; Maneesha Bangar MD[1]; Amira Mohamed MD[1]; Lewis A. Eisen MD[1]; Michael W. Donnino MD[6]

**Affiliations:** 1. Division of Critical Care, Montefiore Medical Center, the Bronx, New York

2. Division of Critical Care, White Plains Hospital, White Plains, New York

3. Quality Management, Montefiore Medical Center, the Bronx, New York

4. Department of Emergency Medicine, Montefiore Medical Center, the Bronx, New York

5. Division of Pulmonary Medicine, Montefiore Medical Center, the Bronx, New York

6. Center for Resuscitation Science, Beth Israel Deaconess Medical Center, Boston, Massachusetts

## Author Contributions

**Conceptualization:** Ari Moskowitz, Xianhong Xie, Michelle Ng Gong, Henry E. Wang, Luke Andrea, Mimi Kim.

**Data curation:** Ari Moskowitz, Henry E. Wang, Yungtai Lo, Mimi Kim.

**Formal analysis:** Xianhong Xie, Mimi Kim.

**Funding acquisition:** Ari Moskowitz.

**Investigation:** Yungtai Lo.

**Methodology:** Ari Moskowitz, Xianhong Xie, Michelle Ng Gong, Luke Andrea, Yungtai Lo.

**Writing – original draft:** Ari Moskowitz, Mimi Kim.

**Writing – review & editing:** Ari Moskowitz, Xianhong Xie, Michelle Ng Gong, Henry E. Wang, Luke Andrea, Yungtai Lo, Mimi Kim.

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
