## [Decision Letter · Decision Letter 0]

8 May 2024

PONE-D-24-07606Exploration of Alive-and-Ventilator Free Days as an Outcome Measure for Clinical Trials of Resuscitative InterventionsPLOS ONE

Dear Dr. Moskowitz,

Thank you for submitting your manuscript to PLOS ONE. After careful consideration, we feel that it has merit but does not fully meet PLOS ONE’s publication criteria as it currently stands. Therefore, we invite you to submit a revised version of the manuscript that addresses the points raised during the review process.

We look forward to receiving your revised manuscript.

Kind regards,

Sebastian Schnaubelt, MD, PhD

Academic Editor

PLOS ONE

Journal Requirements:

Dr. Moskowitz is funded by the National Heart, Lung, and Blood Institute (R33HL162980).

3. One of the noted authors is a group or consortium. In addition to naming the author group, please list the individual authors and affiliations within this group in the acknowledgments section of your manuscript. Please also indicate clearly a lead author for this group along with a contact email address.

Reviewers' comments:

Reviewer's Responses to Questions

**Comments to the Author**

1. Is the manuscript technically sound, and do the data support the conclusions?

Reviewer #1: No

Reviewer #2: No

2. Has the statistical analysis been performed appropriately and rigorously? 

Reviewer #1: I Don't Know

Reviewer #2: I Don't Know

3. Have the authors made all data underlying the findings in their manuscript fully available?

Reviewer #1: Yes

Reviewer #2: Yes

4. Is the manuscript presented in an intelligible fashion and written in standard English?

Reviewer #1: Yes

Reviewer #2: Yes

5. Review Comments to the Author

**Reviewer #1: **This study describes alive-and-ventilator free days (AVFD) as outcome parameter for cardiac arrest trials. The authors try to compare the results of different definitions of AVFD with different results of simulation studies and also with results of an already completed trial. However, the description of some very important points were inadequate or confusing for me. I came away with too many questions to be able to recommend this paper without major revision.

- Please revise the Methods section, especially the part describing the simulation of results. I am not sure in which way the results of the observational cohort, the bootstrapped samples from PART and the PART study influenced the simulation.

- I am very confused of Figure 1 - as this is the observational cohort, what do you mean with intervention and placebo? Try to improve the graphic in general to illustrate the results in a better way.

- In an paper about CPR outcome the Utstein style should be mentioned.

- Please also include in the definition of AVFD what is considered as ventilator. Is a patient for example with NIV ventilator free? Is there a consensus on this in the research community?

**Reviewer #2: **The authors of the present study describe the possible benefits of the composite outcome of alive-and-ventilator-free days in resuscitation trials. They assess different definitions of AVFD as well as different methods for comparison in simulated data. Finally, the evaluated methods are presented on actual data of a completed trial.

However, some points are unclear and confusing to me:

• I do not fully understand the use of the Two-Part modelling framework. As mentioned on page 15 lines 141-146 the first part assesses death/survival or ROSC/death/survival, however these metrics cannot be inferred by the composite AVFD endpoint alone (as you mention yourself on page 13 lines 105-110). Utilizing additional data by one analysis only renders comparability questionable.

o If a composite endpoint for the three levels ROSC/death/survival is desired why didn`t you impose penalty on both unfavorable outcomes (e.g. no ROSC =-2, death after ROSC=-1)?

• Regarding the interpretation of the statistical results it should be noted that the AVFD in the observational cohort clearly does not follow a normal distribution. Did you generate normally distributed AVFD values in the simulation process? If not, what did the used distributions look like?

• Despite the primary focus on the simulation results, the underlying patient collective should be described in a formal and transparent manner to enable assessment of applicability in different cohorts.

• Because of the primary focus on simulated data, the sampling/simulation process should be described in more detail in my opinion.

• Graphics: Please use a different type of format or higher resolution for the graphics. Interpretability in the current form is limited.

• Regarding Table 1:

o Please add a description of the presented values. It is not clear to me if the AVFD days are presented as mean+SD or median+IQR for example.

o Additionally, the presentation differs between the columns. Please ensure uniform presentation.

• Regarding Table 2:

o Please add a description of the presented values. It is not clear to me for example what the values in the LT and ETI column in the first row mean? (I presume mean days+SD?)

o It do not understand the meaning of “Mean proportion of AVFD out of 30 days among survivors: ###%”. This metric is not used or described before.

o Furthermore, the format of the table makes it rather hard to read.

6. PLOS authors have the option to publish the peer review history of their article (what does this mean?). If published, this will include your full peer review and any attached files.

Reviewer #1: **Yes: **Christoph Veigl

Reviewer #2: No

---

## [Author Response · Author response to Decision Letter 0]

7 Jun 2024

Response to Reviewers

Reviewer #1 (R1): This study describes alive-and-ventilator free days (AVFD) as outcome parameter for cardiac arrest trials. The authors try to compare the results of different definitions of AVFD with different results of simulation studies and also with results of an already completed trial. However, the descriptions of some very important points were inadequate or confusing for me. I came away with too many questions to be able to recommend this paper without major revision.

Authors (A): We thank the Reviewer for their comments and have made Major Revisions to the manuscript as suggested by the Reviewers.

R1: Please revise the Methods section, especially the part describing the simulation of results. I am not sure in which way the results of the observational cohort, the bootstrapped samples from PART and the PART study influenced the simulation.

A: We agree that the flow of analyses described in the manuscript may be challenging for a reader to follow. We have substantially revised the Methods section to more clearly show the specific steps taken in the simulation studies. 

As now indicated in the section on study rationale and overall approach, two separate simulations were performed: “model-based simulations” in which data are simulated from an underlying statistical model with parameters informed by results from an observational IHCA cohort, and a second set of “bootstrap” simulations where the simulated data were drawn with replacement from the PART trial data. This approach was taken so as to provide the reader with simulation results when using an in-hospital cardiac arrest AVFD distribution as well as an out-of-hospital cardiac arrest AVFD distribution.

R1: I am very confused of Figure 1 - as this is the observational cohort, what do you mean with intervention and placebo? Try to improve the graphic in general to illustrate the results in a better way.

A: Thank you for this important observation. The Figure 1 attached with the manuscript was incorrect and has been revised in the resubmission. Further, the Figure has been updated for clarity as suggested.

R1: In an paper about CPR outcome the Utstein style should be mentioned.

A: This has been noted as suggested.

R1: Please also include in the definition of AVFD what is considered as ventilator. Is a patient for example with NIV ventilator free? Is there a consensus on this in the research community?

A: This is an important point for the more general PLOS One audience. Ventilation for the purpose of AVFD calculation refers to the administration of positive pressure through an endotracheal or tracheostomy tube. Non-invasive ventilation would not lead to a ventilator day. There is consensus on this definition in the critical care research community.

Reviewer #2 (R2): The authors of the present study describe the possible benefits of the composite outcome of alive-and-ventilator-free days in resuscitation trials. They assess different definitions of AVFD as well as different methods for comparison in simulated data. Finally, the evaluated methods are presented on actual data of a completed trial.

A: Thank you for your thoughtful review.

R2: I do not fully understand the use of the Two-Part modelling framework. As mentioned on page 15 lines 141-146 the first part assesses death/survival or ROSC/death/survival, however these metrics cannot be inferred by the composite AVFD endpoint alone (as you mention yourself on page 13 lines 105-110). Utilizing additional data by one analysis only renders comparability questionable.

A: Thank you for this comment and we apologize that our prior description of the Two-Part method was unclear. We have revised the paper to provide more justification for using the two-part modeling framework to analyze the composite AVFD outcome; we emphasize that modeling in two parts the intervention effect on ROSC/death/survival and the effect on ventilator days among survivors offers greater flexibility and can yield more insights about the relationship between the intervention and individual components of AVFD than standard methods. This is especially true when the effects are in opposite directions across AVFD components, as in Scenario 3 in our simulations (positive effect on reducing risk of ROSC but negative effect on ventilator free days among survivors). We showed that in this case, power to detect an overall intervention effect is greater with the two-part method than with the more restrictive proportional odds model that assumes the intervention effect is the same across all levels of AVFD. With a compositive outcome like AVFD, it is important to consider approaches that allow for greater flexibility in modeling the treatment effect across categories of the outcome. Various types of two-part models exist and have been applied in a number of different fields (PMID 28890906).

R2: If a composite endpoint for the three levels ROSC/death/survival is desired why didn`t you impose penalty on both unfavorable outcomes (e.g. no ROSC =-2, death after ROSC=-1)?

A: This is a very important point and the Reviewer is correct that additional levels for other unfavorable outcomes could have been added. We elected to only study one alternative approach to the AVFD definition to maintain simplicity and because there is precedence from published, high-impact trials for assigning a -1 to a less favorable outcome when analyzing AVFD. We have added these considerations to the Discussion section.

R2: Regarding the interpretation of the statistical results it should be noted that the AVFD in the observational cohort clearly does not follow a normal distribution. Did you generate normally distributed AVFD values in the simulation process? If not, what did the used distributions look like?

A: The Reviewer is correct that AVFD does not follow a normal distribution and this has now been noted in the manuscript. In addition, as described in the revised methods section of the paper, the model-based simulated data was generated in two parts to reflect the non-normality of the AVFD data: first, a multinomial logistic regression model was used to randomly generate for each subject their status with respect to failure to achieve ROSC/death after ROSC/survival. For the survivors, a beta-binomial regression model was then used to randomly generate the number of alive and vent free days. The beta-binomial model was selected to model AVFD among survivors since it yielded the best fit to our IHCA observational data that showed a non-normal distribution with an excess number of values at both ends of the distribution. Given the lack of normality in the AVFD distribution, non-parametric analysis approaches to analyze the AVFD data were considered as part of this study and indeed the findings suggest that use of parametric tests such as a t-test would yield less statistical power. Figure 1 has also been substantially revised to reflect the underlying distribution of the data. Additional details about how the data were simulated are provided in the data supplement.

R2: Despite the primary focus on the simulation results, the underlying patient collective should be described in a formal and transparent manner to enable assessment of applicability in different cohorts.

A: This is an important point. We have added details regarding the observation cohort in Supplemental Table 1. The details of the PART cohort can be found in the original trial publication. This has been noted in the manuscript.

R2: Because of the primary focus on simulated data, the sampling/simulation process should be described in more detail in my opinion.

A: We agree with the Reviewer and have added an extended, technical description of the sampling and simulation process in the Data Supplement.

R2: Graphics: Please use a different type of format or higher resolution for the graphics. Interpretability in the current form is limited.

A: The labeling and resolution of the figures has been improved as suggested.

R2: Regarding Table 1:

o Please add a description of the presented values. It is not clear to me if the AVFD days are presented as mean+SD or median+IQR for example.

o Additionally, the presentation differs between the columns. Please ensure uniform presentation.

A: These changes have been made as suggested.

R2: Regarding Table 2:

o Please add a description of the presented values. It is not clear to me for example what the values in the LT and ETI column in the first row mean? (I presume mean days+SD?)

o It do not understand the meaning of “Mean proportion of AVFD out of 30 days among survivors: ###%”. This metric is not used or described before.

o Furthermore, the format of the table makes it rather hard to read.

A: We agree with the Reviewers assessment and the Table has been modified as suggested.

---

## [Decision Letter · Decision Letter 1]

17 Jul 2024

Exploration of Alive-and-Ventilator Free Days as an Outcome Measure for Clinical Trials of Resuscitative Interventions

PONE-D-24-07606R1

Dear Dr. Moskowitz,

We’re pleased to inform you that your manuscript has been judged scientifically suitable for publication and will be formally accepted for publication once it meets all outstanding technical requirements.

Kind regards,

Sebastian Schnaubelt, MD, PhD

Academic Editor

PLOS ONE

Additional Editor Comments (optional):

Reviewers' comments:

Reviewer's Responses to Questions

**Comments to the Author**

1. If the authors have adequately addressed your comments raised in a previous round of review and you feel that this manuscript is now acceptable for publication, you may indicate that here to bypass the “Comments to the Author” section, enter your conflict of interest statement in the “Confidential to Editor” section, and submit your "Accept" recommendation.

Reviewer #1: All comments have been addressed

Reviewer #2: All comments have been addressed

2. Is the manuscript technically sound, and do the data support the conclusions?

Reviewer #1: Yes

Reviewer #2: (No Response)

3. Has the statistical analysis been performed appropriately and rigorously? 

Reviewer #1: Yes

Reviewer #2: (No Response)

4. Have the authors made all data underlying the findings in their manuscript fully available?

Reviewer #1: Yes

Reviewer #2: (No Response)

5. Is the manuscript presented in an intelligible fashion and written in standard English?

Reviewer #1: Yes

Reviewer #2: (No Response)

6. Review Comments to the Author

Reviewer #1: Thank you for addressing all our comments. Please also add the programs and softwares you used to conduct the study

Reviewer #2: (No Response)

7. PLOS authors have the option to publish the peer review history of their article (what does this mean?). If published, this will include your full peer review and any attached files.

Reviewer #1: **Yes: **Christoph Veigl

Reviewer #2: No

---

## [Editor Report · Acceptance letter]

22 Jul 2024

PONE-D-24-07606R1 

PLOS ONE

Dear Dr. Moskowitz, 

I'm pleased to inform you that your manuscript has been deemed suitable for publication in PLOS ONE. Congratulations! Your manuscript is now being handed over to our production team.

Kind regards, 

on behalf of

Dr. Sebastian Schnaubelt 

Academic Editor

PLOS ONE